# NorQuAD: Norwegian Question Answering Dataset

**Sardana Ivanova,**[1] **Fredrik Aas Andreassen,**[2] **Matias Jentoft,**[2]
**Sondre Wold**,[2] and **Lilja Øvrelid**[2]

[1] University of Helsinki, Department of Computer Science
[2]University of Oslo, Language Technology Group
sardana.ivanova@helsinki.fi
{fredaan, matiasj, sondrewo, liljao}@ifi.uio.no

## Abstract

In this paper, we present NorQuAD: the first Norwegian question answering dataset for machine reading comprehension. The dataset consists of 4,752 manually created question-answer pairs. We detail the data collection procedure and present statistics about the dataset. We also benchmark several multilingual and Norwegian monolingual language models on the dataset and compare them against human performance. The dataset will be made freely available.[1]

## 1 Introduction

Machine reading comprehension is one of the key problems in natural language understanding. The question answering (QA) task requires a machine to read and comprehend a given text passage, and then answer questions about the passage. In recent years, considerable progress has been made toward reading comprehension and question answering for English and several other languages (Rogers et al., 2022).

In this paper, we present NorQuAD: the first Norwegian question answering dataset for machine reading comprehension. The dataset consists of 4,752 question-answer pairs manually created by two university students. The pairs are constructed for the task of extractive question answering aimed at probing machine reading comprehension (as opposed to information-seeking purposes), following the methodology developed for the SQuAD-datasets (Rajpurkar et al., 2016, 2018). The creation of this dataset is an important step for Norwegian natural language processing, considering the importance and popularity of reading comprehension and question answering tasks in the NLP community.

In the following we detail the dataset creation (section 3), where we describe the passage selection and question-answer generation, present relevant statistics for the dataset and provide an analysis of human performance including sources of disagreement. In order to further evaluate the dataset as a benchmark for machine reading comprehension, we perform experiments (section 4) comparing several pre-trained language models, both multilingual and monolingual models, in the task of question-answering. We also compare models against human performance for the same task. We further provide an analysis of performance across the source data domain and annotation time and present the results of manual error analysis on a data sample.

## 2 Related Work

Cambazoglu et al. (2021) categorise QA datasets into abstractive, extractive, and retrieval-based. In *abstractive* datasets the answer is generated in free form without necessarily relying on the text of the question or the document. In *extractive* datasets the answer needs to be a part of a given document that contains an answer to the question. In *retrieval-based* QA, the goal is to select an answer to a given question by ranking a number of short text segments (Cambazoglu et al., 2021). Since NorQuAD is constructed based on extractive QA, we will here concentrate on related work in extractive QA.

The Stanford Question Answering Dataset (SQuAD) 1.1 (Rajpurkar et al., 2016) along with SQuAD 2.0 (Rajpurkar et al., 2018) which supplements the dataset with unanswerable questions are the largest extractive QA datasets for English. SQuAD 1.1 contains 100,000+ questions and SQuAD 2.0 contains 50,000 questions.

Several SQuAD-like datasets exist for other languages. The French Question Answering Dataset (FQuAD) is a French Native Reading Compre-

---

[1]https://github.com/ltgoslo/NorQuAD

hension dataset of questions and answers on a set of Wikipedia articles that consists of 25,000+ samples for version 1.0 and 60,000+ samples for version 1.1 (d'Hoffschmidt et al., 2020). The German GermanQuAD is a dataset consisting of 13,722 question-answer pairs created from the German counterpart of the English Wikipedia articles used in SQuAD (Möller et al., 2021). The Japanese Question Answering Dataset (JaQuAD) consists of 39,696 question-answer pairs from Japanese Wikipedia articles (So et al., 2022). The Korean Question Answering Dataset (KorQuAD) consists of 70,000+ human-generated question-answer pairs on Korean Wikipedia articles (Lim et al., 2019). The Russian SberQuAD consists of 50,000 training examples, 15,000 development, and 25,000 testing examples (Efimov et al., 2020)[2]. To the best of our knowledge there are no extractive question answering datasets available for the other Nordic languages, i.e., Danish or Swedish.

## 3 Dataset Creation

We collected our dataset in three stages: (i) selecting text passages, (ii) collecting question-answer pairs for those passages, and (iii) human validation of (a subset of) created question-answer pairs. In the following, we will present these stages in more detail and provide some statistics for the resulting dataset as well an analysis of disagreements during human validation.

### 3.1 Selection of passages

Rogers et al. (2020) reported that the absolute majority of available QA datasets target only one domain with rare exceptions. To provide some source variation in our dataset, considering our limited resources, we decided to create question-answer pairs from passages in two domains: Wikipedia articles and news articles.

We sampled 872 articles from Norwegian Bokmål Wikipedia. In order to include high-quality articles, we sampled 130 articles from the 'Recommended' section and 139 from the 'Featured' section. The remaining 603 articles were randomly sampled from the remaining Wikipedia corpus. From the sampled articles, we chose only the "Introduction" sections to be selected as passages for annotation. Following the methodology proposed for the QuAIL dataset (Rogers et al.,

2020) with the goal of making the dataset more complex, we selected articles with "Introduction" sections containing at least 300 words.

For the news category, we sampled 1000 articles from the Norsk Aviskorpus (NAK)—a collection of Norwegian news texts[3] for the year 2019. As was the case with Wikipedia articles, we chose only news articles which consisted of at least 300 words.

### 3.2 Collection of question answer-pairs

Two students of the Master's program in Natural Language Processing at the University of Oslo, both native Norwegian speakers, created question-answer pairs from the collected passages. Each student received separate set of passages for annotation. The students received financial remuneration for their efforts and are co-authors of this paper. For annotation, we used the Haystack annotation tool [4] which was designed for QA collection. An example from the Haystack annotation environment for a Norwegian Wikipedia passage is shown in Figure 1. The annotation tool supports the creation of questions, along with span-based marking of the answer for a given passage. In total, the annotators processed 353 passages from Wikipedia and 403 passages from news, creating a total of 4,752 question-answer pairs. The remaining collected passages could be used for further question-answer pair creation.

#### 3.2.1 Instructions for the annotators

The annotators were provided with a set of initial instructions, largely based on those for similar datasets, in particular, the English SQuAD dataset (Rajpurkar et al., 2016) and the German-QuAD data (Möller et al., 2021). These instructions were subsequently refined following regular meetings with the annotation team. The annotation instructions will be made available along with the dataset.

#### 3.2.2 Question generation

Annotators were instructed to read the presented passages and formulate 5-10 questions for each passage. The questions should be varied in terms of wh-question type: *hva* 'what', *hvor* 'where', *når* 'when', *hvem* 'who', *hvilke* 'which', *hvordan*

---

[2]The datasets are presented in alphabetical order

[3]https://www.nb.no/sprakbanken/en/
resource-catalogue/oai-nb-no-sbr-4/

[4]https://github.com/deepset-ai/
haystack/

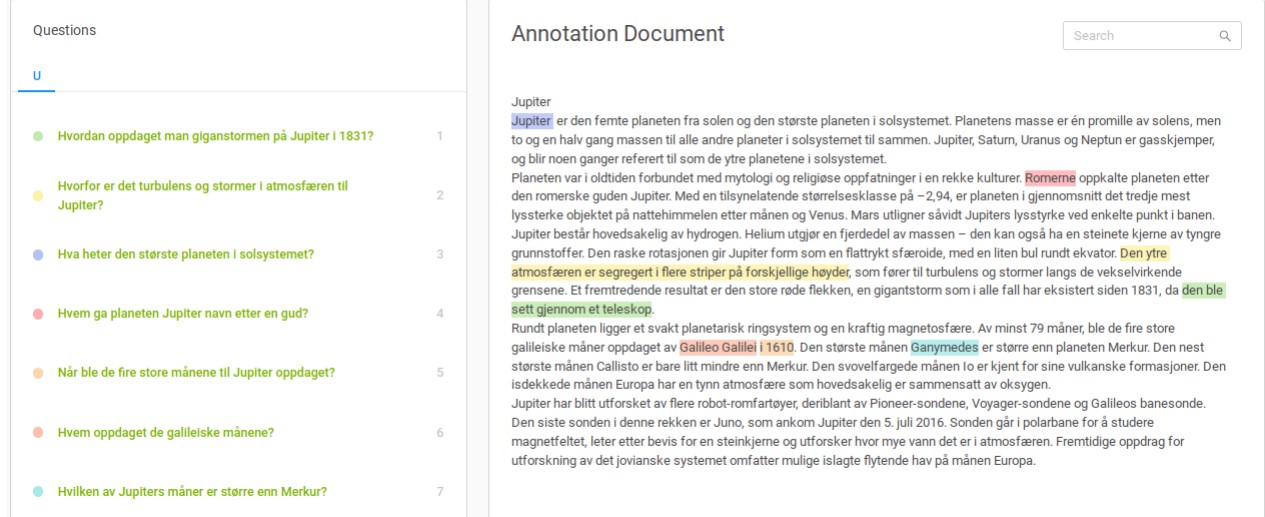

Figure 1: View of the Haystack annotation environment for a Norwegian Wikipedia document. The tool supports the creation of questions along with span-based marking of the selected answer for a document.

'how' and *hvorfor* 'why'. When formulating questions, the annotators were further instructed not to repeat or simply copy words or phrases from the passage text directly, but rather, if possible, rephrase the question. They were provided with a number of examples of types of re-phrasals, inspired by the Japanese QA dataset JaQuAD (So et al., 2022):

**Syntactic variation** The questions should, if possible, make use of syntactic alternations, such as the active-passive alternation:
  *... John Lennon was assassinated by Mark Chapman on ..*
  Q: Who assassinated John Lennon?

**Lexical variation (synonymy)** The questions should if possible make use of synonymy relations in re-phrasal:
  *... John Lennon was assassinated by Mark Chapman on ..*
  Q: Who murdered John Lennon?

**Lexical variation (inference)** The questions should if possible make use of inference based on lexical or world knowledge in re-phrasal:
  *... John Lennon was assassinated by Mark Chapman on December 8, 1980 ...*
  Q: When did John Lennon die?

**Multiple sentence reasoning** The questions should if possible require inference based on more than one sentence in the associated passage:
  *... John Lennon was the world-famous guitarist of The Beatles. He wrote many songs, among them "All you need is love".*
  Q: Who wrote "All you need is love"?

In general, the annotators were encouraged to pose difficult questions as long as they can be answered based on the information in the passage (and additional inference). The questions should in combination cover most of the passage, however, if this turned out to be difficult to balance with the requirement to pose varied questions, a priority should be given to the latter requirement. Each question should have only one answer and there are no unanswerable questions in the dataset.

### 3.2.3 Answer generation

The annotators were instructed to mark answers to their questions that adhere to the following main principles:

- The answer should consist of the shortest span in the original passage that answers the question.

- The answer should, however, also be a natural-sounding and a grammatically correct response to the question. As an example, for the question "When was Lennon born?" the answer text span should include the preposition "in" and not only the year "1940" if "in 1940" is indeed a span of the original text.

| Question word | Wikipedia | News | Total |
|---|---|---|---|
| *hva* 'what' | 507 (21.54%) | 383 (15.97%) | 890 (18.73%) |
| *hvor* 'where' | 414 (17.59%) | 471 (19.64%) | 885 (18.62%) |
| *når* 'when' | 381 (16.19%) | 385 (16.06%) | 766 (16.12%) |
| *hvem* 'who' | 350 (14.87%) | 393 (16.39%) | 743 (15.64%) |
| *hvilke* 'which' | 346 (14.70%) | 325 (13.55%) | 671 (14.12%) |
| *hvordan* 'how' | 201 (8.54%) | 267 (11.13%) | 468 (9.85%) |
| *hvorfor* 'why' | 152 (6.46%) | 174 (7.26%) | 326 (6.86%) |
| other | 3 (0.13%) | 0 (0%) | 3 (0.06%) |
| Total | 2354 | 2398 | 4752 |

Table 1: Question types distribution by question word in the dataset, broken down by data source (Wikipedia/news).

- Answers should always consist of whole words, and there should be no subword answers, such as parts of a compound or words stripped of affixes.

- Answer spans should furthermore not include span-final punctuation.

- The answers to the question should only occur once in the passage. Sometimes the same entity occurs multiple times, but it should occur only once as an answer to the relevant question.

### 3.3 Dataset analysis

To understand the properties of the created question-answer pairs, we automatically categorised the whole NorQuAD dataset by question word. We provide statistics for the questions and their distribution by question word in Table 1. The "other" row in the table contains questions which we could not automatically categorise by a question word due to absence of a question word in a question or a typo in a question word. The table shows that the distribution of the various question types is fairly balanced, with the most common type being *hva* 'what' type questions (18.73% of all questions) and the least common being *hvorfor* 'why' type questions (6.86%). While the annotators were instructed to try to introduce variation in question types, the distribution of these will depend on the type of data. There are clear differences between the two data sources (Wikipedia and news), and we find that the news data contains more *hvor* 'where', *hvem* 'who' and *hvordan* 'how' type questions and less *hva* 'what' type questions than the Wikipedia portion of the

| Question word | Wikipedia | News | Total |
|---|---|---|---|
| *hva* 'what' | 136 | 123 | 259 |
| *hvor* 'where' | 107 | 177 | 284 |
| *når* 'when' | 121 | 100 | 221 |
| *hvem* 'who' | 84 | 104 | 188 |
| *hvilke* 'which' | 95 | 88 | 183 |
| *hvordan* 'how' | 61 | 89 | 150 |
| *hvorfor* 'why' | 46 | 47 | 93 |
| Total | 650 | 728 | 1378 |

Table 2: Question types distribution for human validation

dataset.

The reason for a lower occurrence of *hvorfor* 'why' and *hvordan* 'how' type questions is related to this dataset being extractive in its nature. For Norwegian, these question words require answers of a particular form, which do not occur as frequently in descriptive text as in other types of language data, such as dialogue.

It is worth noting that the overview in Table 1 does not differentiate distinct question types for questions using *hvor* as an adverb of degree, e.g. *hvor mange* 'how many', *hvor ofte* 'how often' and *hvor gammel* 'how old'. Even though *hvor* in these questions does not denote location, they are categorized as *hvor* 'where' type questions, in contrast to the GermanQuAD dataset, where *wie viele* 'how many' questions are considered a separate question type rather than just *wie* 'how' type questions (Möller et al., 2021). We found that 214 (24.18%) of our total 885 questions categorised as *hvor* 'where' type questions, are actually questions asking *hvor mange* 'how many'.

| Question word | Wikipedia | | News | | Total | |
|---|---|---|---|---|---|---|
| | EM | F1 | EM | F1 | EM | F1 |
| *hva* 'what' | 66.2% | 85.8% | 74.0% | 91.7% | 69.9% | 88.6% |
| *hvor* 'where' | 82.2% | **94.7%** | 85.9% | 94.9% | 84.5% | 94.8% |
| *når* 'when' | 81.8% | 92.2% | **94.0%** | 97.5% | **87.3%** | 94.6% |
| *hvem* 'who' | 73.8% | 88.8% | 83.7% | 93.3% | 79.3% | 91.3% |
| *hvilke* 'which' | 65.3% | 89.1% | 68.2% | 88.5% | 66.7% | 88.8% |
| *hvordan* 'how' | 72.1% | 87.0% | 89.9% | 92.7% | 82.7% | 90.4% |
| *hvorfor* 'why' | **82.6%** | 90.8% | 91.5% | **99.0%** | 87.1% | **94.9%** |
| Total | 74.3% | 89.8% | 83.4% | 93.7% | 79.1% | 91.8% |

Table 3: Averaged human performance by question types

## 3.4 Dataset validation

In a separate stage, the annotators validated a subset of the NorQuAD dataset. In this phase each annotator replied to the questions created by the other annotator. We chose the question-answer pairs for validation at random. In total, 1378 questions from the set of question-answer pairs, were answered by validators. This provides us with a measure of human performance on a subset of the dataset. Table 2 shows the number of question-answer pairs assessed by a human, as broken down by the question types over the two data sources. It is worth mentioning that this subset is larger than the test set for which modeling results are reported in later sections. Table 3 shows the performance of the human annotators in terms of exact match and token-level F1. For the dataset as a whole we find a human performance of 79.1% exact match and a token-level F1 of 91.8%.

Exact match measures the percentage of predictions that exactly match the ground truth answers. F1 score is the harmonic mean of precision and recall. We calculate token-level F1. Both metrics ignore punctuation.

**Wikipedia** The annotators answered a total of 650 questions taken from the Wikipedia category. We found that human performance is 74.3% for the exact match metric and 89.8% for token-wise F1 score on average. These results are further broken down by question types in Table 3. We find that *hvilke* 'which' type questions seem to be the most difficult, with an exact match score of only 65.3%, however with considerably higher F1, indicating that for this category the precise delimitation of the answer span proves challenging. The question type with the highest F1 score of 94.7% is the *hvor* 'where', most likely due to location

expressions being relatively easy to identify. The question type with the highest EM score of 82.6% is *hvorfor* 'why'.

**News** The annotators answered a total of 728 questions taken from the news category of the dataset. Overall exact match for this source is 83.4% with a total F1 of 93.7%. Somewhat surprisingly, the human results for this category turned out to be overall higher than for the Wikipedia category. For the news category, the *hvilken* 'which' type questions have the lowest human performance (EM 68.2%; F1 88.5%) which proved difficult also for the Wikipedia articles. The question type with the highest scores in terms of human performance for the news category are the *når* 'when' and *hvorfor* 'why' type questions.

## 3.5 Analysis of disagreement

As presented in Table 3, we found that human validators performed better on the news dataset than on the Wikipedia dataset. One possible reason for this is that the time of annotation affected the quality of question-answer pairs seeing that the news dataset was annotated after Wikipedia annotation. We explore this hypothesis further in Section 4.3 below. Here we examine the disagreements between the annotators during the validation phase in some more detail.

A manual inspection of the disagreements shows that there are primarily three categories of disagreements between the annotator and the validator, two of which are semantic and the last one grammatical in nature. First, there is disagreement caused by the decision of how big of a span is necessary to answer a question fully. For example, for the question *Hva er ei runebomme laget av?* 'What is a runebomme (type of drum) made

from?', the answer could be either *et dyreskinn stort sett reinskinn* 'an animal skin mainly from reindeer skin' or *et dyreskinn stort sett reinskinn spent over en oval treramme eller over en oval uthult rirkule* 'an animal skin mainly from reindeer skin, pulled over an oval wooden frame or over an oval hollowed out burl'. The first option excludes the description of how the reindeer skin is constructed, while the second one includes it.

The second category of disagreement is where an all together different span in the text is selected to answer the question. In these cases, the question is either not precisely enough formulated by the annotator to exclude other options, or the validator has been semantically imprecise in their understanding of the question. For the cases of the first type, ideally there should be no ambiguity. For example, for the question 'Where was Napoleon born?', the answer could be explicit one place in the text, as in '...He was born in Corsica..', and implicit in another '...born 15 August 1769, Corsica'. Both alternatives for the string '(in) Corsica' would be valid options, so in this case the problem is an imprecisely formulated question. For the question 'In what city is Nidarosdomen?', the answer 'Trøndelag' would be wrong, as that is a region and not a city. In that case the mistake is on the validator side, as the only correct answer would be 'Trondheim'.

The last category has to do with function words like determiners and prepositions, and whether to repeat them in the answer. Here the principles of answering with the shortest possible span but at the same time ensuring that the answers are natural sounding are in conflict. One example containing the subjunction *som* 'as' is the answer to the question *Hva arbeidet Miklos Horthy **som** fram til 1944?* 'What did MH work **as** until 1944?': *som statsoverhode i ungarn* 'as head of state in Hungary' versus the alternative answer span *statsoverhode i ungarn* which excludes the subjunction.

## 4   Experiments

In this section, we assess the use of the NorQuAD dataset as a benchmark for Norwegian machine reading comprehension. Given the small size of the dataset, compared to many other QA datasets, one important question to assess will be the level of performance that can be obtained with less than 5,000 question-answer pairs.

To evaluate models, we use two metrics which are used to evaluate performance on most SQuAD-like datasets: exact match and F1 score, as described in Section 3.4.

We split the NorQuAD dataset randomly into three sets: training (80%), validation (10%), and test (10%). We split datasets to the above-mentioned fractions separately for Wikipedia and news datasets to observe if performance will differ depending on domain. The test sets consist of human validated question-answer pairs, hence we may compare models' results to human performance on the same data. The human validation process is described in Section 3.4.

We establish benchmarking experiments using a set of different pre-trained language models, outlined below. Due to the small size of NorQuAD, as compared to other SQuAD-like datasets, we run all configurations five times with different random seeds and report the mean and standard deviation from these experiments. Details on selected hyperparameters are located in Appendix A.1

### 4.1   Baseline models

**Norwegian models.**   We compare two monolingual transformer models based on the architecture from BERT (Devlin et al., 2019): The NorBERT2, originating from the initiative started in NorLM (Kutuzov et al., 2021), and the NB-BERT model from Kummervold et al. (2021).

**Multilingual models.**   We further compare two multilingual models: mBERT (Devlin et al., 2019) and XLM-RoBERTa (Conneau et al., 2020).

**Cross-lingual augmentation.**   Table 1 shows that $\approx 51\%$ of the questions in our dataset use the question words *what, where* and *who*. Such questions are often answered with a named entity. Hence, we hypothesise that a lot of the annotated gold labels are named entities, and that much of the performance on SQuAD-like datasets therefore comes down to identifying the span of the correct entity in the text, which is less language specific. To investigate this, we study whether or not we can utilise another SQuAD-like dataset as a cross-lingual data augmentation step in order to increase the number of samples.

We warmup the NB-BERT model on the GermanQuAD (Möller et al., 2021) dataset for 3000 optimization steps with a batch size of 16 using a learning rate of $1e - 4$. That is, we first fine-tune the NB-BERT model on the German data for a

| Model | Wiki | | News | | All | |
|---|---|---|---|---|---|---|
| | EM | F1 | EM | F1 | EM | F1 |
| Human* | 72.65 | 88.84 | 83.61 | 93.43 | 78.13 | 91.14 |
| NorBERT2 | 57.76 ± 1.15 | 71.89 ± 0.89 | 64.05 ± 1.27 | 76.93 ± 1.15 | 64.64 ± 1.40 | 77.86 ± 0.65 |
| NB-BERT | 59.74 ± 0.76 | 74.16 ± 1.31 | 67.64 ± 1.11 | 79.17 ± 0.92 | **69.68** ± 1.21 | **81.27** ± 0.73 |
| mBERT | 55.70 ± 1.67 | 71.21 ± 1.20 | 63.12 ± 2.34 | 73.96 ± 1.26 | 63.32 ± 1.58 | 76.00 ± 0.83 |
| XLM-RoBERTa | 54.33 ± 7.14 | 70.00 ± 7.76 | 61.72 ± 2.74 | 75.62 ± 2.88 | 64.52 ± 1.37 | 78.42 ± 0.97 |
| NB-BERT$_{ger}$ | **65.23** ± 1.45 | **78.504** ± 1.67 | **70.80** ± 1.59 | **80.76** ± 1.78 | 68.78 ± 1.38 | 80.76 ± 0.62 |

Table 4: Results on the test set of the different domains of the NorQuAD dataset. Results are reported as means over five different random seeds with standard deviation. *Human performance is the averaged performance of the two annotators on complementary halves of the test set (10%). The model NB-BERT$_{ger}$ refers to the model with the cross-lingual augmentation from the GermanQuAD dataset. Note that the full human validated part of the dataset is larger, hence these results are not identical to those reported in Table 3.

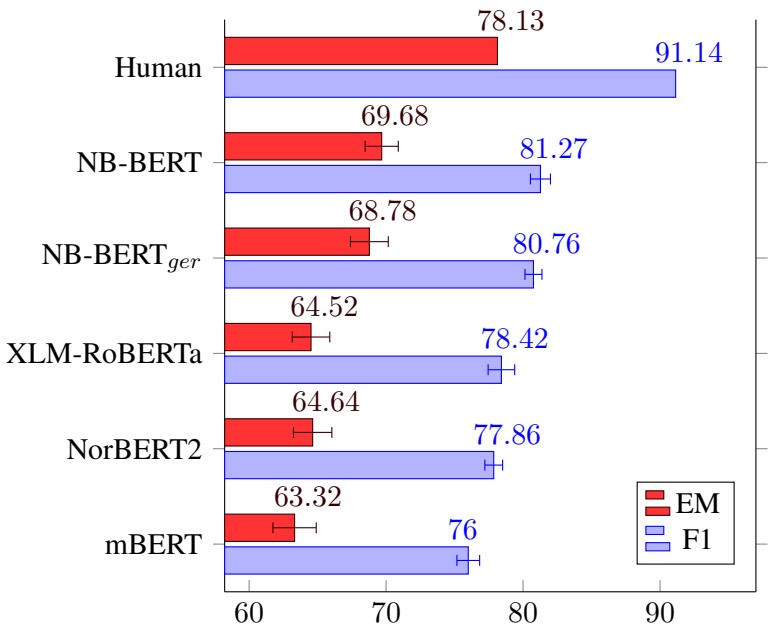

Figure 2: Performance of annotators and models on the entire test set: EM and F1 scores

limited number of steps, then switch to the Norwegian data and continue fine-tuning in the same way as for the monolingual experiments. We choose German as the augmentation language because it is typologically similar to Norwegian and since the GermanQuAD dataset has a similar annotation scheme. We chose NB-BERT as the model for cross-lingual augmentation because it was the best performing monolingual model.

## 4.2 Results

Table 4 shows the results of the baseline models outlined above. Overall, all models perform worse on the Wikipedia split, followed by the news split, and perform best on the total split. The best performing monolingual model, NB-BERT, performs better than both multilingual models and also has

the lowest standard deviation over the different runs. We note that the performance of XLM-RoBERTa is particularly unstable. As this architecture is very similar to the models based on BERT, we did not perform a separate round of hyperparameter tuning for this model, which might explain the instability. A comparison of performance of models against human performance on all the data (both Wikipedia and news) is shown in Figure 2.

Our results indicate that it is possible to further improve the performance of a monolingual model by first warming up on the GermanQuAD dataset. This model achieves the highest score out of all the baselines on both the Wikipedia and news split with an exact match score of 65.23% and 70.80%. However, on the total split the cross-lingual data

augmentation step did not yield any added performance and performed on par with just regular fine-tuning, which points towards the cross lingual warmup being most effective for low sample scenarios. That is, when there is enough training data available, the models converge towards the same point regardless of the warmup phase. Furthermore, as the multilingual models also perform close to the monolingual ones, we interpret this as evidence towards the importance of identifying named entities for closed question answering.

Although the models perform well with respect to the relatively low sample size, as compared to other SQuAD-like datasets, there is still room for improvement when considering the human performance level.

### 4.3 Performance across domains and time

We noticed a difference in performance both for annotators and models on the Wikipedia and news partitions of the data sets, where the annotators and models generally performed better on the news partition. During the data collection phase, the annotators started creating question-answer pairs first for Wikipedia passages and subsequently moved on to news passages. One relevant question is therefore whether the observed difference in performance is an artifact of the way the data collection was organized. It might be that the annotators became more competent at the task over time and that the annotation for the news section is therefore more consistent and generally of a higher quality.

To evaluate whether the time of annotation affected the consistency of annotation, we measure the performance of the annotators (as obtained during the data validation stage) as well as the best performing NB-BERT model on the halves of the test dataset which were created first and the halves which were created last. These partitions measure 117/117 for Wikipedia and 119/119 for news. The results can be observed in Figure 3.

The results show that the annotators perform better on question-answer pairs created later in the annotation process both for the Wikipedia and news partitions of the dataset. We find that the average performance of the annotators on the beginning of the Wikipedia dataset is 67.29% EM and 84.99% F1, compared with 70.35% EM and 91.83% F1 on the last part of this dataset. For the beginning of the news dataset the results are

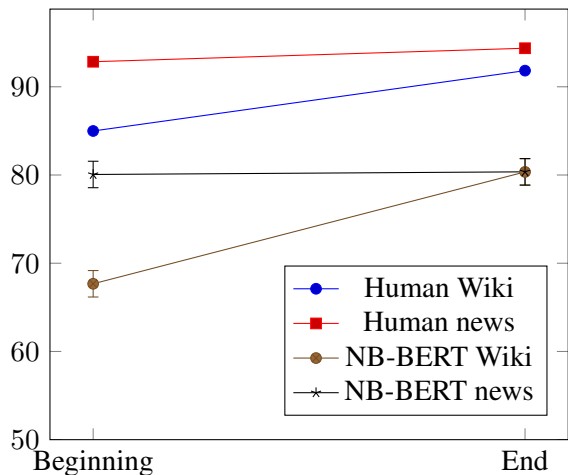

Figure 3: Performance on the beginning and end of datasets (F1 score)

81.79% EM and 92.84% F1, while the last part achieves the higher results of 85.70% EM and 94.37% F1. In total for the beginning half of the whole dataset, the performance is 74.54% EM and 88.92% F1, while the averaged performance of the annotators for the last halves are 78.03% EM and 93.10% F1.

For the models, the average performance over five runs on the beginning of the news dataset is 67.73% EM and 80.06% F1 compared to 71.35% EM and 80.36% F1 at the end. The difference in EM is within a standard deviation. On the Wikipedia dataset, however, the discrepancy is large: the average performance on the beginning is 54.98% EM and 67.67% F1 compared to 66.32% EM and 80.35% F1 at the end, with a standard deviation of $\approx 1.5$ at both tails.

These results indicate a temporal effect in the data collection phase, where the annotators became more consistent in their question-answer pairs over time regardless of the domain. Since they started out with Wikipedia this effect is most noticeable in the Wikipedia category.

## 5 Error Analysis

In this section, we have a more detailed look at how the models performed on our dataset. We sampled 60 errors from the system output. The model in question is NB-BERT. We found that in 57% of cases there is at least some overlap between the prediction and the annotated span. In 85% of instances the predicted answers are grammatically and semantically viable phrases, but are not factually the correct answer to the current

question. Within this category there are cases of questions where the answer is the same type of entity as the one asked for in the question (e.g. a person, but the wrong person), and in other cases the entity is of a different type (e.g. a number, instead of a person).

In 13% of cases the predicted phrase could be considered a good answer to the question, but the annotator made a different selection. This means that the question must be considered ambiguous and that the system is not necessarily to blame for the error, but rather the annotation itself, because according to the annotation guidelines each question should have only one possible answer (see Section 3.2.2). Please see further discussion on this type of ambiguity in section 3.5 on analysis of disagreement between annotators.

## 6   Conclusion

In this paper, we presented NorQuAD—the first question answering dataset for Norwegian. We collected passages from Norwegian Wikipedia articles and a collection of Norwegian news texts and manually created over 4,700 questions. In the experiments, we fine-tuned several pre-trained language models with NorQuAD and found that the best performing model achieved $69.68\%$ for EM and $81.27\%$ for F1 score on the entire test set. Averaged human performance on the test set was $78.13\%$ for EM and $91.14\%$ for F1. The dataset and our experiments are available at `https://github.com/ltgoslo/NorQuAD`. Furthermore, we presented human validation of the part of the dataset.

We noticed that annotators and models performed better on the news dataset and we performed experiments to find out whether the time of annotation influences the performance. We measured performance of annotators and NB-BERT—the best performing model. We found that both annotators and the model perform better on the second half of the datasets meaning that the annotators got better and more consistent in their question-answer pairs over time.

While it is clear from our experiments that the number of question-answer pairs are sufficient to achieve a decent performance, improvements are certainly possible both in terms of size and data quality. To improve the dataset quality, one possible avenue for future work is to collect multiple answers for the questions as was done in

SQuAD (Rajpurkar et al., 2016). Another possible extension in the future is the addition of unanswerable questions (Rajpurkar et al., 2018) to the dataset.

## Acknowledgements

The project was funded by a grant from Teksthub of the University of Oslo. The project was supported in part by scholarship from Erasmus traineeship program (SMP). We are grateful for three anonymous reviewers for their many insightful comments and suggestions.

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

# A   Appendix

## A.1   Hyperparameters

The hyperparameters for the baseline models were selected based on a grid search against the validation split. The final parameters were used for all models on all splits, except for the warmup phase of the cross lingual augmentation configuration.

- $lr = 5e - 5$

- $epochs = 3$

- $batch\_size\_train = 16$

- $batch\_size\_eval = 8$

- $learning\_rate\_scheduler = linear$