# OpenReview forum: "NorQuAD: Norwegian Question Answering Dataset"
_NoDaLiDa/2023/Conference — NoDaLiDa 2023_

### Official Review · Reviewer_3Zgh · 2023-02-27
**A good paper with thorough description of the data collection and a strong evaluation part.**

**Rating:** 8
**Confidence:** 4

**Review:**

This paper describes the construction of NorQuAD, a relatively small Norwegian Question-Answering dataset, for machine reading comprehension. Two annotators (students) generated about 5,000 question-answer pairs using text passages collected from two sources. Additionally, the paper presents the results of an evaluation of several different models trained on NorQuAD.

Overall, the description of the data collection is thorough and clear.  Question types distribution for both data sources is presented which is important information.  A subset of the data was validated by making an annotator marking answers to questions that the other annotator had previously marked. The agreement between the annotators was used as a measure of human performance on this subset.

In the evaluation part, both Norwegian language models and multilingual models were fine-tuned on NorQuAD and compared with human performance.  Temporal effects were analysed and cross-lingual data augmentation was applied.

Given that this paper is a student paper, the work described seems quite impressive.  The work is not very original, but it is very significant for the Nordic LT community and overall the text is clear. There are some issues that indeed need to be clarified and/or fixed (see the list below), but those issues are relatively easy to fix.

Citations and bibliography
------------------------------
All entries in the bibliography need to be checked for correct title casing. One example is:
Qa dataset explosion: A taxonomy of nlp resources for question answering and reading compre- 926 hension -> QA Dataset Explosion: A Taxonomy of NLP Resources for Question Answering and Reading Comprehension

Some entries are incomplete, e.g. "Pranav Rajpurkar, Robin Jia, and Percy Liang. 2018. 918 Know what you don’t know: Unanswerable questions for squad" has missing Journal info.

"In recent years considerable progress has been made toward reading comprehension and question answering for English and several other languages."  You need to provide some references to support this claim.

"and the NB-BERT model from (Kummervold et al., 2021)" -> "and the NB-BERT model by Kummervold et al. (2021)"

Clarifications/fixing needed
-------------------------------
Both the Abstract and the Introduction should contain a brief description of the main results of the paper.

Title case should be used for main section headings (but not for subsections).  For example, "Related work" -> "Related Work"

"Each question should have only one answer and there are no unanswerable questions in the dataset".  You should elaborate on this point - why did you decide not to posy any unanswerable questions?

"with the most common type being hvor ‘what’ type question" -> "with the most common type being hva ‘what’ question"" (Note: not 'hvor', but rather 'hva')

"In total, 1378 questions from the set of question-answer pairs, were answered by a human."  Sounds as if the remaining questions were not answered by a human!  I assume that you mean "answered by the validator".

Table 1 is too far away from the text that refers to it.  You need to move this table to the top of page 4.  Same goes for Tables 2 and 3.

"Table 3 shows the performance of the human annotators in terms of exact match and token-level F1."  You should define what "exact match and "token-level" means immediately after this sentence (your definition comes in the next paragraph and only for exact match)

"We warmup the NB-BERT model on the GermanQuAD (Möller et al., 2021) dataset for 3000 optimization steps".  What is the purpose of the "warm-up"?

"Fine-tuning is then done as normal on NorQuAD" ->  You should tell the reader what is the "normal way".  Moreover, which library/framework did you use for the training/fine-tuning/testing?

"... and it is also the most stable across seeds." -> You should tell the reader what is meant by "stable" (i.e. refer to the standard deviation)

The caption for Table 4 should remind the reader of the difference between NB-BERT and NB-BERT-ger

If this paper is accepted, you will have one additional page and I therefore suggest that you add a table showing the performance figures for the beginning and end datasets (currently listed in the text below Figure 3).

"This means that the question must be considered ambiguous, and that the error is not necessarily one of the system, but of the annotation itself." Not clear enough - I suggest:
"This means that the question must be considered ambiguous and that the system is not necessarily to blame for the error, but rather the annotation itself, because according to the annotation guidelines each question should have only one possible answer (see Section 3.2.2)"

"This provides us with a measure of human performance on a subset of the dataset."  Was this (i.e. using validation statistics as a measure of human performance) your idea or did you find this in some previously published work?


Spelling, grammar and minor issues
-----------------------------------------
Abstract:
"In this paper we present NorQuAD" -> "In this paper, we present NorQuAD"

"We here detail the data collection procedure and present statistics of the dataset." -> "We detail the data collection procedure and present statistics about the dataset."

Introduction:
"In recent years considerable progress has been made toward ... " -> "In recent years, considerable progress has been made toward ..."

Related Work:
"The largest extractive QA dataset for English is Stanford Question Answering Dataset ..." -> "the Stanford Question Answering Dataset ..."

"There are several SQuAD-like datasets for other languages ..." -> "Several SQuAD-like datasets exist for other languages ..."

"French Question Answering Dataset (FQuAD) is ..." -> "The French Question Answering Dataset (FQuAD) is ..."

"... of the English Wikipedia articles’ used in SQuAD" -> "... of the English Wikipedia articles used in SQuAD"

"... consists of 39,696 question-answer pairs on Japanese Wikipedia articles" -> "... from Japanese Wikipedia articles"

"Russian SberQuAD consists of 50,000 training ..." -> "The Russian SberQuAD consists of 50,000 training ..."


Dataset Creation:
"We collect our dataset in three stages" -> "We collected our dataset in three stages"

"In order to include high-quality articles we sampled 130 articles ..." -> "In order to include high-quality articles, we sampled 130 articles ..."

"From the sampled articles we chose only ..." -> "From the sampled articles, we chose only ..."

"As with Wikipedia articles we chose only news articles ..." -> "As was the case with Wikipedia articles, we chose only news articles ..."

"Instructions to the annotators" -> "Instructions for the annotators"

"The questions should if possible make use ..." -> ""The questions should, if possible, make use ..."

"The answer should, however, also be a natural-sounding and grammatical response to the question" -> "The answer should, however, also be a natural-sounding and a grammatically correct response to the question"

"We find that the distribution of the various question types are fairly balanced," -> "The table shows that the distribution of the various question types is fairly balanced,"

"For Norwegian these question words ..." -> "For Norwegian, these question words ..."

"We find that 214 (24.18 %) of our total ..." -> "We found that 214 (24.18%) of our total ..."

"In a separate stage, the annotators validate a subset of the NorQuAD dataset ..." -> "validated"

"We find that human performance is 74.3%" -> "We found that human performance is 74.3%"

"As presented in Table 3 we found that ..." -> "As presented in Table 3, we found that "

"We here examine ..." -> "Here we examine ..."

"If the question is Hva er ei runebomme laget av? ..." -> "For example, for the question Hva er ei runebomme laget av? ..."


Experiments:
"In the following we assess the use of the NorQuAD dataset ..." -> "In this section, we assess the use of the NorQuAD dataset ...

"To evaluate models we use two metrics which are used to evaluate performance on most SQuAD-like datasets: exact match and F1 score, as described above." ->
"To evaluate models, we use two metrics which are used to evaluate performance on most SQuAD-like datasets: exact match and F1 score, as described in Section 3.4."

"Details on selected hyperparameters are located in appendix A.1" -> "Appendix A.1"

Error Analysis:+
Use past tense throughout, e.g. "we sampled" and we found" instead of "we sample" and "we find"

"We have a more detailed look at how the models performed on our dataset." -> "In this section, we have a more detailed look at how the models performed on our dataset.


Conclusion:
"In this paper, we presented NorQuAD—the first question answering dataset for Norwegian." -> "In this paper, we presented NorQuAD — the first question answering dataset for Norwegian."

"Besides, we presented human validation of the part of the dataset." -> "Furthermore, we presented human validation of the part of the dataset."

"Furthermore, we noticed that annotators ..." -> "We noticed that annotators ..."


**Paper Type:**

Long paper

---

### Official Review · Reviewer_ms9v · 2023-03-07
**The paper presents the new question-answer dataset for the Norwegian language and some experiments with it.**

**Rating:** 7
**Confidence:** 4

**Review:**

The paper presents the new question-answer dataset for the Norwegian language and some experiments with it. My remarks:
1.	2 annotators had to create question-answer pairs. It is not clear if they got separate passages. If they got the same, how have you assured that question-answer pairs do not overlap? Besides, despite the provided instructions each annotator can have a bit different understanding of how question-answer pairs have to be created (e.g., one may prefer longer answers, etc.).
2.	You state that the annotators were instructed to paraphrase questions (formulated from the passage texts). Have you investigated how well did they follow these instructions?
3.	What do F1 and token-level F1 values in Table 3 really demonstrate? Do they demonstrate some quality of answers or simply the fact that the same question can be correctly answered in several ways (longer/shorter span; with some additional function words, etc.)?
4.	The manual inspection of disagreements (from line 464) emphasizes three categories, but does not provide any statistics on how disagreements are distributed among them (which of these categories is the most/least common). The second category is when the different span in the text is selected. Is this problem related to the incorrectly formulated question or to the incorrectly selected answer?
5.	Table 4: NB-BERT_germ seems the best option if having Wiki and News texts separately, but having both is not. Please, explain this phenomenon. Maybe you needed to run each experiment more than 5 to see the real picture. Are you using the same training and testing subsets in All as in Wiki and News separately?
6.	Is there any reason why only NB_BERT was selected for the cross-lingual experiments? It would be very interesting to know how the larger training dataset (including the German language) could impact the results with other methods (e.g., XLM-RoBERTa).
7.	It is difficult objectively evaluate the performance of automatic methods: the answers can be presented differently, but still remain correct.

**Paper Type:**

Long paper

---

### Official Review · Reviewer_TQEE · 2023-03-08
**A nice presentation of a Norwegian question answering data set**

**Rating:** 7
**Confidence:** 3

**Review:**

This paper presents a newly developed Norwegian QuAD question answering data set. It describes how the data set was collected and evaluates it using several pre-trained language models.Overall, the paper contains a nice description which is easy to read.

A few questions for clarification:

- In related work, it may be good to also mention the availability of QuADs, or similar data sets, for related languages, such as Swedish and Danish.
- In 3.2 the passages processed were about half of the collected articles. Were the others discarded or do you hope to use them for creating more question-answer pairs?
- In 3.2.3 you mention subtoken answers. Could you define what this means?
- In 3.3 you say that a variation for question types between 6 and 18% is "fairly balanced". Would it have been possible/desirable to have it fully balanced?
- In 3.5 you give examples for most disagreements, except for the differences in span. While there are examples later, maybe it would make it easier for the reader to give one here as well.
- In section 4.3 the results are reported again as numbers. Could you maybe discuss and compare instead of just reporting the numbers? It's a bit difficult to read.

Some minor spelling etc comments:

- If possible try to place the tables and figures close to where they are mentioned.
- Line 480: the sentence should probably not end with a question mark.
- Figure 2 is difficult to read in black and white. Could you choose a darker colour for one of the two scores?
- Line 840: validation of a part of

**Paper Type:**

Long paper

---

### Decision · Program_Chairs · 2023-03-17

Accept